# Machine learning for syndromic surveillance using veterinary necropsy reports

**Nathan Bollig**[1,2]*, **Lorelei Clarke**[3], **Elizabeth Elsmo**[3], **Mark Craven**[1,4]

**1** Department of Computer Sciences, University of Wisconsin-Madison, Madison, WI, United States of America, **2** Department of Pathobiological Sciences, School of Veterinary Medicine, University of Wisconsin-Madison, Madison, WI, United States of America, **3** Wisconsin Veterinary Diagnostic Laboratory, University of Wisconsin-Madison, Madison, WI, United States of America, **4** Department of Biostatistics and Medical Informatics, University of Wisconsin-Madison, Madison, WI, United States of America

* nbollig@wisc.edu

**Data Availability Statement:** Data cannot be shared publication due to the free-text laboratory reports. The Wisconsin Veterinary Diagnostic Laboratory (WVDL) has an ethical responsibility to protect the confidentiality of test results and

## Abstract

The use of natural language data for animal population surveillance represents a valuable opportunity to gather information about potential disease outbreaks, emerging zoonotic diseases, or bioterrorism threats. In this study, we evaluate machine learning methods for conducting syndromic surveillance using free-text veterinary necropsy reports. We train a system to detect if a necropsy report from the Wisconsin Veterinary Diagnostic Laboratory contains evidence of gastrointestinal, respiratory, or urinary pathology. We evaluate the performance of several machine learning algorithms including deep learning with a long short-term memory network. Although no single algorithm was superior, random forest using feature vectors of TF-IDF statistics ranked among the top-performing models with F1 scores of 0.923 (gastrointestinal), 0.960 (respiratory), and 0.888 (urinary). This model was applied to over 33,000 necropsy reports and was used to describe temporal and spatial features of diseases within a 14-year period, exposing epidemiological trends and detecting a potential focus of gastrointestinal disease from a single submitting producer in the fall of 2016.

## Introduction

More than 60% of emerging infectious diseases can be transmitted from animals, making animal populations an important surveillance tool for detecting emerging disease [1]. Because animals share the same environment as humans and often spend more time outdoors, they are also important for monitoring environmental health hazards, human health hazards, and bioterrorism threats [2].

While there is a growing emphasis on monitoring data captured early in the course of medical evaluation or treatment, such as clinical notes or lab request forms (often called pre-diagnosis data), existing animal disease surveillance systems frequently depend on definitive diagnoses achieved through lab testing [3,4]. Such systems exhibit a time delay in detecting novel or unexpected diseases emerging in a population and may exhibit poor sensitivity to multifactorial diseases that cannot be characterized by a single agent [5]. Surveillance relying on pre-diagnosis data targets broad categories of diseases and is often called "syndromic

reports. In accordance with this ethical responsibility and the AAVLD requirement, WVDL policy allows for a researcher to contact a laboratory staff member or quality manager (Kristin Zuzek, 1521 E. Guy Ave, Barron WI, 54812). The researcher should complete a Third Party Information Request Form, which will then be forwarded to the WVDL director (Keith Poulsen, 445 Easterday Lane, Madison WI, 53706) for final review.

**Funding:** Funding was provided by the Comparative Biomedical Sciences Training Program (NIH T32OD010423) and the Wisconsin Veterinary Diagnostic Laboratory. The Wisconsin Veterinary Diagnostic Laboratory provided support for the study in the form of salaries for authors LC, EE and in the form of the article publication fee. MC was supported by NIH/NCATS UL1 TR000427. The funders had no role in study design, data collection and analysis, decision to publish, or preparation of the manuscript.

**Competing interests:** The authors have declared that no competing interests exist.

surveillance" [6]. By facilitating the rapid detection of potential public and animal health threats, syndromic surveillance can enable the implementation of targeted investigations, diagnostic testing, or prophylactic treatments early in the course of a potential outbreak.

Necropsies are post-mortem evaluations performed by veterinarians in the field and at diagnostic facilities to determine the cause of an animal's illness or death, and are often critical in the investigation of disease outbreaks in a herd [7]. Necropsy reports represent a unique opportunity for syndromic surveillance because of their emphasis on an animal's cause of death, and because the text is often dominated by specific morphologic terms describing grossly observable and microscopic tissue changes. The reports also commonly include the animal's signalment, clinical signs, geographic origin, and herd-level factors [8].

As is common for pre-diagnosis data, necropsy reports are often written in a free-text format. Analysis of free text is generally challenging, and natural language processing (NLP) methods have become increasingly important in mining clinical text [9]. Text mining can be used to classify passages into categories, such as disease groups, which may be monitored for changes over time. This framework has been used to conduct syndromic surveillance using chief complaints in human records [10,11]. In animals, text mining has been used to conduct syndromic surveillance from online news reports [12], web searches [13,14], and laboratory test requests [4].

There is a growing amount of literature examining information-extraction tasks involving pathology reports [15–21]. A rule-based approach is common, in which prediction rules are manually built, commonly using pre-defined named entities recognized using NLP software. Such algorithms often suffer from the knowledge acquisition bottleneck associated with maintaining extensive lists of named entities and the rules governing their interpretation, resulting in a loss of portability and flexibility [4,22,23]. A rule-based text mining system for syndromic surveillance has been recently described in the context of veterinary necropsy reports [8].

Machine learning does not require the manual development of decision rules as it automatically infers a model from an annotated corpus. While supervised machine learning requires human input to produce document labels, this approach is generally less intensive than designing and maintaining a set of rules [23]. Machine learning has been successfully used to extract a multitude of discrete phenotypes from heterogenous health data including free text [24,25]. Current literature represents a variety of learning algorithms useful for medical text analysis [26] and multiple approaches to encoding document features including n-gram ("bag of words") representations [25], graphs-of-words [24,27,28], and sequential encodings with deep learning [29–33]. Current reports indicate that recurrent neural network (RNN) models such as long short-term memory (LSTM) networks [34] can be highly successful for veterinary text classification tasks when a large amount of training data is available [35]. They are also reported as effective models for syndromic surveillance using free-text chief complaints in human medicine [10].

We aim to demonstrate that supervised machine learning methods can effectively perform syndromic classification of free-text veterinary necropsy reports, forming the basis for an automated approach to syndromic surveillance within an animal population. We focus on evaluating distinct machine learning algorithms and show that some are effective for this task. We also demonstrate that a preliminary predictive signal can be extracted from gross necropsy findings alone, which approximately represents the first available information in a necropsy examination.

## Methods

### Data

Necropsy reports were obtained from the Wisconsin Veterinary Diagnostic Laboratory (WVDL) at the University of Wisconsin-Madison. Necropsy submissions at this facility

represent most species of veterinary importance with a strong emphasis on farm animals, particularly bovine. All electronic necropsy reports on record between July 6, 2004 and August 6, 2018 were acquired as raw data for a total of 33,567 reports. Each necropsy report included five sections: (1) *gross necropsy findings*, (2) *histological findings*, (3) *morphologic findings*, (4) *final diagnosis*, and (5) *pathologist comments*. The reports also included additional information such as the animal receipt date, location, species, breed, and sex.

## Construction of a document dataset for labeling

Using the R Programming Language [36], a subset of 1,000 reports was randomly sampled from the dataset. For each pathology report, a *primary document* was prepared by combining the *morphologic findings* and *final diagnosis* sections or, if both of those were empty, by combining all sections (15% of cases). Since the most concise morphologic terminology is present in these sections, this abstraction submitted only the most structured language to the learning model.

## Defining syndromes

Because a necropsy examination is organized according to organ systems in the animal, we selected examples of topographical, organ-system-based syndromic categories: (1) gastrointestinal (GI) disease, (2) respiratory disease, and (3) urinary disease. These categories were intentionally general and inclusive of both overt and non-specific illnesses relating to each respective system. For example, documents describing evidence of diarrheal disease or non-specific hepatic disease should both be flagged as positive by a GI-disease classifier. To illustrate the language in WVDL pathology reports, Table 1 presents examples of text criteria judged by two veterinary pathologists to represent positive classifications in each syndromic category.

## Obtaining expert labels

Two veterinarians board-certified by the American College of Veterinary Pathologists reviewed the 1,000 documents and classified each as having evidence of GI disease and/or

**Table 1. Necropsy text examples.**

| Syndrome | Phrases |
|---|---|
| **Gastrointestinal Disease** | • Moderate, acute suppurative enteritis<br>• Abomasum: Submucosal hemorrhage<br>• Intestine: There are clusters of necrotic cells present within crypts and the mucosa is congested<br>• Necrosuppurative rumenitis, widely disseminated, marked, acute<br>• Moderate, acute, non-granulocytic portal hepatitis |
| **Respiratory Disease** | • Multifocal to coalescing, moderate, acute aspiration pneumonia<br>• Suppurative bronchopneumonia<br>• Multifocal to coalescing alveolar atelectasis<br>• Interstitial congestion and edema with intra alveolar meconium<br>• Scattered bronchiolar necrosis |
| **Urinary Disease** | • Subacute nephrosis with granular and hemoglobin casts<br>• Diffuse tubular nephrosis; multifocal, subacute interstitial nephritis<br>• Hydronephrosis, severe<br>• Membranous glomerulonephritis<br>• Bilateral lymphoplasmacytic pyelitis |

Phrases from Wisconsin Veterinary Diagnostic Laboratory (WVDL) necropsy reports that were judged by two veterinary pathologists as representing gastrointestinal, respiratory, or urinary pathology.

respiratory disease and/or urinary disease based on clinical experience. Diagnoses were excluded that did not specify an organ system, such as "salmonellosis", "bacteremia", or "septicemia". A small percentage of randomly selected documents (3%) were blank and classified as negative in all three syndromic categories. The inter-rater reliability between the two experts was measured using percent agreement and Cohen's kappa. One expert was selected to represent ground-truth syndrome labels.

## Defining the machine learning task

The machine learning model should evaluate a necropsy report and determine if there is evidence of GI, respiratory, or urinary pathology. Any, all, or none of these syndromes could be present. This was accomplished by developing a separate binary classifier for each syndrome. A document was fully processed after being independently evaluated by each of the classifiers, an approach generally useful for multi-label classification in medical record prediction tasks [37]. This allows for learned models to be customized to each syndrome and would allow the pipeline to be augmented with additional classifiers later without affecting the pre-existing steps.

## Statistical analysis of performance

The performance of a binary classifier can be evaluated by its accuracy:

$$\text{Accuracy} = \frac{\text{Correct predictions}}{\text{Total predictions}}$$

However, accuracy is not ideal for studying classification performance in cases where positive instances of a syndrome are significantly over- or underrepresented in the training data. To make our analysis robust to class skew, we also utilized the following metrics for each binary classifier:

$$\text{Recall} = \frac{\text{True positives}}{\text{True positives} + \text{False negatives}}$$

$$\text{Precision} = \frac{\text{True positives}}{\text{True positives} + \text{False positives}}$$

These metrics were combined using a harmonic mean into a single performance metric called the F1 score:

$$\text{F1} = \frac{2}{\frac{1}{\text{recall}} + \frac{1}{\text{precision}}} = 2 \cdot \frac{\text{precision} \cdot \text{recall}}{\text{precision} + \text{recall}}$$

In this study, performance statistics were reported using 10-fold cross-validation, and 95% confidence intervals were computed using bootstrapping as described in Gao et al. [29,38] and summarized in Table 2. All references to statistical significance are made relative to a significance level of 5%.

## Learning with bag of words representations

Document text was tokenized into words and cast into a document term matrix (DTM) (Figs 1 and 2). In this process, each document was separated into a collection of words, reflecting a bag-of-words approach that does not preserve the original order of document terms. The DTM is a large, sparse matrix in which each row represents a document and each column

**Table 2. Determining confidence intervals in cross-validation experiments.**

| | |
|---|---|
| **Step 1:** **Cross-Validation** | Pool test set predictions across cross-validation folds. |
| **Step 2:** **Bootstrapping** | Repeat 2000 times: <br> • Sample with replacement from the pooled predictions to create a bootstrapped set of predicted labels equal in size to the set of pooled predictions. <br> • Calculate the F1 score of the classifier using this bootstrapped set. |
| **Step 3:** **Confidence Interval Calculation** | Determine the 2.5 and 97.5 percentile of the distribution of F1 scores computed in Step 2. |

represents a unique word in the document corpus. Columns corresponding to the common pathology terms "mild", "moderate", "acute", "multifocal", "small", "diffuse", and "necrosis" were removed because they could be used in reference to any body system and are therefore not relevant for syndromic prediction. Stop words were also removed from consideration.

Each entry in the DTM encodes the term frequency–inverse document frequency (TF-IDF) measure for the corresponding document and word. Term frequency (TF) measures how frequently the word appears in the document. I.e. if $n_{ij}$ represents the number of times term $t_i$ appears in document $d_j$ then the frequency of term $t_i$ in document $d_j$ is

$$\text{TF} = \frac{n_{ij}}{\text{Total number of terms in } d_j}.$$

The following expression gives the inverse document frequency (IDF) of term $t_i$:

$$\text{IDF} = \log_2 \frac{\text{Total number of documents}}{\text{Number of documents containing } t_i}$$

The IDF of a term provides a weight inversely correlated to its frequency across all text. Finally,

*Examined is the body of a 2.0 pound, 8.0 ounce 29.0 cm male Holstein fetus with no gross lesions.*

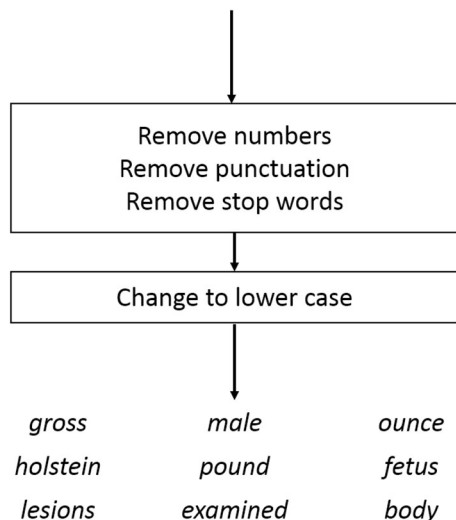

**Fig 1. Tokenization.** A document example was tokenized into words. Numbers and punctuation were removed. Stop words, common words in English (like "from", "and", and "of") were removed. All characters were changed to lower case. After tokenization, the document was represented as a non-ordered collection of words.

|  | 1<br>abdomen | 2<br>abdominal | ... | $p$ - 1<br>zones | $p$<br>zooepidemicus |
|---|---|---|---|---|---|
| 1 | $x_{1,1}$ | $x_{1,2}$ | ... | $x_{1,\,p-1}$ | $x_{1,\,p}$ |
| 2 | $x_{2,1}$ | $x_{2,2}$ | ... | $x_{2,\,p-1}$ | $x_{2,\,p}$ |
| ⋮ | ⋮ | ⋮ | ... | ⋮ | ⋮ |
| 1000 | $x_{1000,1}$ | $x_{1000,2}$ | ... | $x_{1000,\,p-1}$ | $x_{1000,\,p}$ |

**Fig 2. Document term matrix.** After all documents were tokenized, the results were summarized in a document term matrix (DTM). There were $p$ unique words extracted in the tokenization process, with several examples shown. Each row represents a document, and each column represents a word. Entry $x_{ij}$ in row $i$, column $j$ represents the term frequency–inverse document frequency (TF-IDF) for the $j$-th term in the $i$-th document. The DTM is a sparse matrix in which most entries are zero.

the TF-IDF score for term $t_i$ in document $d_j$ is the product

$$\mathrm{TF-IDF} = \mathrm{TF} \cdot \mathrm{IDF}.$$

This approach encodes each document as a feature vector of TF-IDF statistics. Using this representation, we evaluated the performance of several machine learning methods on the syndromic classification task defined above. Models were learned using scikit-learn [39] in Python version 3.7. On each cross-validation fold, the hyperparameter space was explored using a grid search, and hyperparameters were selected to maximize mean F1 scores computed by internal 10-fold cross-validation. To assess feature importance weights in tree-based models, the normalized mean decrease in Gini impurity was summarized using scikit-learn.

**Logistic regression.** Logistic regression with L2 regularization was evaluated with cost parameters in $\{10^{-4}, 10^{-3}, \ldots, 10^{4}\}$.

**Support vector machine.** A support vector machine aims to find a hyperplane separating documents in feature space [40]. A grid search was performed to consider both linear and Gaussian radial basis function kernels, cost in $\{2^{-5}, 2^{-3}, \ldots, 2^{15}\}$, and gamma in $\{2^{-15}, 2^{-13}, \ldots, 2^{3}\}$.

**Classification and regression tree (CART).** An optimized CART algorithm was evaluated using the standard decision tree model in scikit-learn. The maximal depth of the tree was controlled by specifying the minimum number of documents $min_{samples}$ required to split an internal node. Values of this hyperparameter in the set {2, 5, 10, 50} were considered.

**Bagging trees.** Bagging (bootstrap aggregation) represents a statistical ensembling technique in which each tree is trained on documents sampled randomly with replacement [41]. This was done using 1,000 trees learned via the CART method on each cross-validation fold. All trees had $min_{samples}$ globally fixed to the value selected by internal cross-validation when using CART to learn single trees, so that no hyperparameter searching was employed for this algorithm.

**Random forest.** A random forest is another tree-based ensemble learner in which bootstrapped sampling is applied and a random subset of features is considered to produce the split at each node of every decision tree [42]. Each model used 1,000 trees. The depth of each tree was controlled using $min_{samples}$ as in the CART model, and the maximum number of features considered for each node split was specified as a hyperparameter $m$. Given a feature space of size $p$, grid search considered $m$ in the set

$$\{\sqrt{p}, 0.02p, 0.05p, 0.1p, 0.2p, 0.5p, 0.75p, 0.9p, 0.95p, 1.0p\}$$

(with values rounded down to the nearest integer) and $min_{samples}$ in {2, 5, 10, 50}.

**Gradient tree boosting.** We also considered gradient tree boosting, in which shallow decision trees are iteratively combined into a stronger ensemble learner [43]. Each model used 1,000 boosting stages. A grid search explored maximum tree depths in $\{2, 3, \ldots, 10\}$ and learning rates in $\{10^{-5}, 10^{-4}, \ldots, 10^{-1}, 1\}$.

## Deep learning with sequence representations

Document text was encoded using a 50,000-word vocabulary. Accordingly, each document was represented by a sequence of integers uniquely determined by the sequence of words in the text (Fig 3). These sequences were padded to a maximum length of 50 words. Keras [44] and TensorFlow [45] in Python were used for text pre-processing and model implementation.

A recurrent neural network model was considered for the syndromic classification task (Fig 4). When propagating a document forward through the network, each vocabulary word was first projected into a 200-dimensional GloVe embedding space in which the Euclidean distance is smaller between pairs of more similar words [46]. After the initial embedding layer, there was a 1-dimensional convolutional layer consisting of 64 3x1 filters employing ReLU activations, and subsequently a 1-dimensional max pooling operation with a window size of 4 and valid padding. Next there was a single long short-term memory (LSTM) layer with 128 hidden units, followed by a densely-connected, single output unit with a sigmoid activation function. To prevent overfitting, dropout [47] was used between the embedding and convolutional layers, and L2 regularization was employed at the convolutional and output layers. The model was trained using Adam optimization with its default parameters [48], binary cross-entropy loss, and a mini-batch size of 32 over 10 epochs. The matrix of embedding parameters was initialized using GloVe embeddings but subjected to gradient descent updates throughout training.

## Error analysis

For selected learning methods, we conducted an error analysis to provide human interpretation of model predictions. At the end of cross-validation, predictions on each test fold were concatenated to yield a set of predictions for the entire labeled dataset. This was provided to a human reviewer as a spreadsheet, who attempted to identify and quantify major classes of errors via manual inspection of the input document text.

## Classifying documents beyond the labeled corpus

To demonstrate applications of the syndromic classifiers, we trained the highest-performing model (measured by F1 score) on the entire labeled corpus. A 10-fold cross-validated grid search was employed as in the initial model validation experiments to ensure that hyperparameters were optimal. This model was used to predict GI syndrome classifications on the entire document corpus, which were then used to generate a time-series of GI disease cases in R. Cases involved in a sharp rise in prevalence were examined as a possible disease outbreak. Other examples of analysis specific to the GI syndrome were explored.

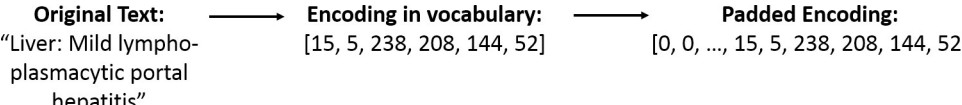

**Fig 3. Document encoding.** Each word was represented by an index pointing to the word's position in a fixed 50,000-word vocabulary. This sequence was padded with zeros to a length of 50.

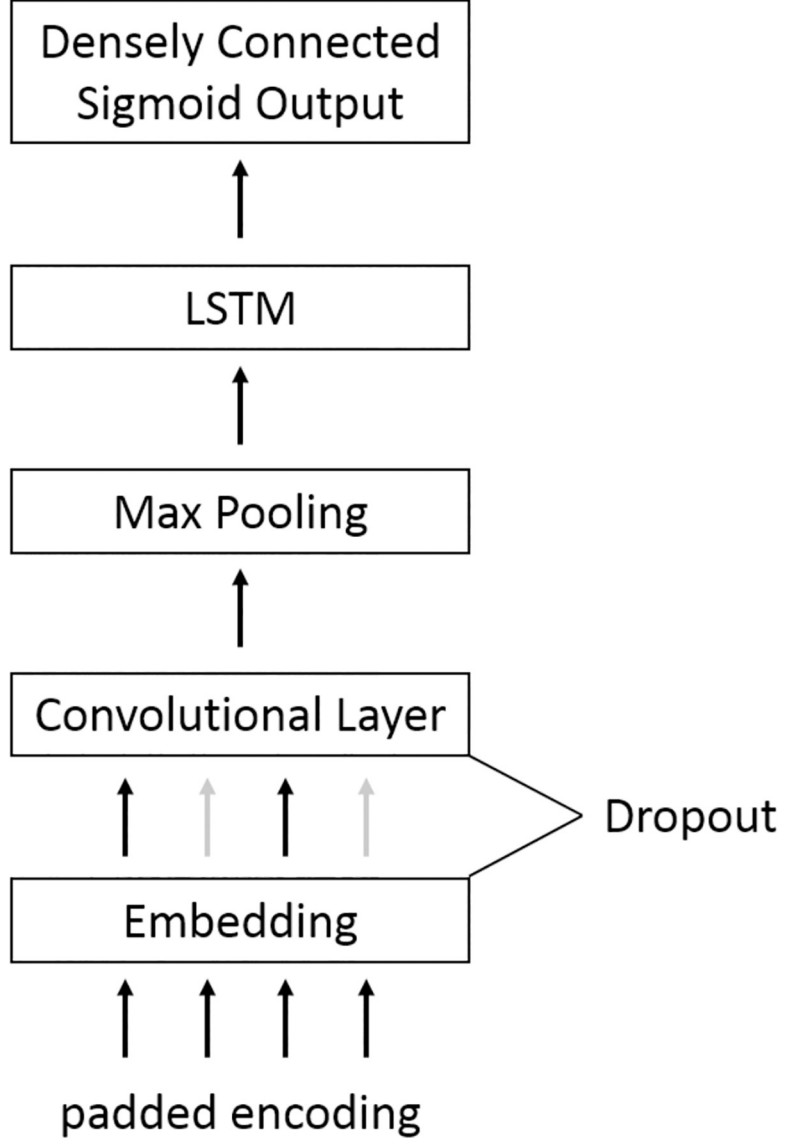

**Fig 4. Long Short-term memory network architecture.**

## Learning from gross necropsy findings

The first section of a WVDL necropsy report (*gross necropsy findings*) is often a valid approximation of the document's initial draft status. Given that all sections of the report relate to the same patient and a single necropsy exam, we hypothesized that the syndrome label assigned to the primary document represents a valid label for gross findings. We applied all methods reported in the section "Learning with bag-of-words representations" except that we tested models on TF-IDF representations of only the *gross necropsy findings* section. Analysis was restricted to the subset of documents for which this section was non-empty. Both primary documents and *gross necropsy findings* were evaluated as training input. For each syndrome, the performance of these learners was compared to a baseline syndromic classifier whose output is indiscriminately positive. F1 scores and 95% confidence intervals were computed using the same bootstrapping procedure.

## Results

Two experts achieved percentage agreement of 97.2% and a Cohen's kappa of 0.944 for their labeling of 1,000 documents from the dataset. After defining one expert's labels as ground truth, a proportion of 51.1% (511/1000) represented the GI syndrome, 45.8% (458/1000) represented respiratory disease, and 10.8% (108/1000) represented urinary disease.

### Learning with bags-of-words representations

Table 3 presents accuracy and F1 scores for machine learning approaches applied to each of the syndromic classification tasks, using unigram TF-IDF vectors as input features. The dimension of the feature space was 2,594. Although no single model was best, random forest was consistently among the top-performing models with F1 scores of 0.923 (GI), 0.960 (respiratory), and 0.888 (urinary). Logistic regression and support vector machine models exhibited lower performance. Precision-recall curves for the random forest model are presented in Fig 5. Optimal hyperparameters are described in Table A in S1 Text. The inclusion of bigram tokens did not significantly improve performance and may cause a marginal performance degradation for these models (Table B in S1 Text).

The mean decrease in Gini impurity provides a static illustration of feature importance for the random forest model, helping to explain which features have the greatest impact on its classification decisions (Table 4).

### Deep learning with sequence representations

F1 scores of 0.932 (GI), 0.947 (respiratory), and 0.752 (urinary) were achieved by the LSTM network (Table 3). Precision-recall curves are presented in Fig 6. The F1 scores of all models are summarized graphically in Fig 7.

### Error analysis

Manual error inspection was performed for the random forest model. False negative predictions outnumber false positives for two of the three syndromic prediction tasks (Fig 8).

**Table 3. Performance metrics for machine learning models.**

|  | GI Disease | | Respiratory Disease | | Urinary Disease | |
|---|---|---|---|---|---|---|
| **Model** | **Accuracy** | **F1** | **Accuracy** | **F1** | **Accuracy** | **F1** |
| Logistic Regression | 0.889 (0.870, 0.908) | 0.893 (0.873, 0.913) | *0.912 (0.894, 0.928)* | *0.904 (0.884, 0.923)* | *0.930 (0.914, 0.945)* | *0.642 (0.562, 0.718)* |
| Support Vector Machine | 0.892 (0.872, 0.910) | 0.892 (0.872, 0.912) | *0.918 (0.901, 0.934)* | *0.909 (0.888, 0.928)* | *0.942 (0.928, 0.955)* | *0.693 (0.612, 0.764)* |
| Classification Tree | 0.899 (0.880, 0.918) | 0.899 (0.879, 0.918) | 0.960 (0.948, 0.972) | 0.956 (0.942, 0.969) | 0.972 (0.961, 0.982) | 0.861 (0.807, 0.906) |
| Bagging Trees | **0.923 (0.907, 0.939)** | **0.923 (0.905, 0.940)** | **0.963 (0.951, 0.974)** | **0.959 (0.946, 0.972)** | 0.975 (0.965, 0.984) | 0.872 (0.821, 0.918) |
| Random Forest | **0.923 (0.906, 0.938)** | **0.923 (0.906, 0.939)** | **0.963 (0.951, 0.974)** | **0.960 (0.947, 0.972)** | **0.978 (0.968, 0.986)** | **0.888 (0.843, 0.930)** |
| Gradient Tree Boosting | 0.903 (0.884, 0.921) | 0.902 (0.882, 0.920) | 0.961 (0.949, 0.972) | 0.957 (0.944, 0.969) | **0.977 (0.967, 0.986)** | **0.887 (0.839, 0.929)** |
| LSTM Network | **0.933 (0.917, 0.949)** | **0.932 (0.917, 0.947)** | 0.952 (0.938, 0.965) | 0.947 (0.931, 0.962) | 0.9539 (0.9400, 0.9670) | 0.752 (0.683, 0.819) |

Three syndrome classification tasks (gastrointestinal, respiratory, and urinary) were tested. Accuracy and F1 scores were assessed by 10-fold cross-validation, with 95% confidence intervals in parentheses calculated by bootstrapping. The two best results in each column are bolded. Results outside the confidence intervals of the best results are italicized.

**Table 4. Feature importance for random forest.**

| GI | | Respiratory | | Urinary | |
|---|---|---|---|---|---|
| **Weight** | **Feature** | **Weight** | **Feature** | **Weight** | **Feature** |
| 0.193 | enteritis | 0.330 | pneumonia | 0.321 | kidney |
| 0.114 | liver | 0.271 | lung | 0.120 | nephritis |
| 0.075 | intestine | 0.159 | bronchopneumonia | 0.116 | lesions |
| 0.052 | hepatitis | 0.041 | pulmonary | 0.090 | renal |
| 0.026 | hepatic | 0.036 | lungs | 0.054 | kidneys |
| 0.023 | abomasitis | 0.028 | lesions | 0.049 | tubular |
| 0.016 | cryptosporidiosis | 0.015 | tracheitis | 0.048 | significant |
| 0.015 | enteric | 0.007 | edema | 0.028 | bladder |
| 0.013 | intestinal | 0.006 | interstitial | 0.020 | urinary |
| 0.012 | cryptosporidia | 0.005 | congestion | 0.016 | hydronephrosis |

The most important features for random forest models trained in each syndromic task. Weights reflect the normalized total decrease in Gini impurity associated with each feature.

False negatives were most frequently associated with an uncommon term, a species-specific anatomical descriptor, or terms derived from causative organisms (Table C in S1 Text). In total, these accounted for 48% (40/84) of false negative predictions. Uncommon terms included references to specific tissues, cell types, or disease processes appearing so infrequently that it seemed unreasonable for a machine learner to recognize their significance without addition domain knowledge (27% of false negatives). Species-specific anatomical descriptors were tracked separately and mostly included descriptors of avian and ruminant anatomy (18% of false negatives). Terms derived from causative organisms were associated with 14% of false negatives. These percentages do not add to 48% because a small number of documents

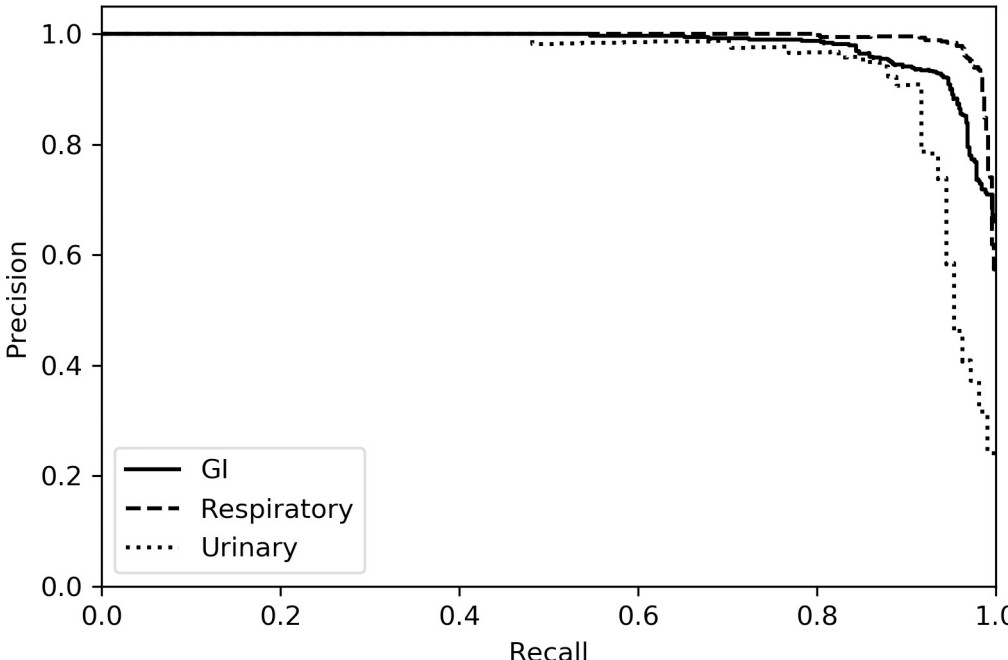

**Fig 5. Precision-recall curves for random forest.** Confidence values for test-set predictions were pooled across cross-validation folds. Areas under the curve (AUCs) are 0.981 (GI), 0.994 (respiratory), and 0.947 (urinary).

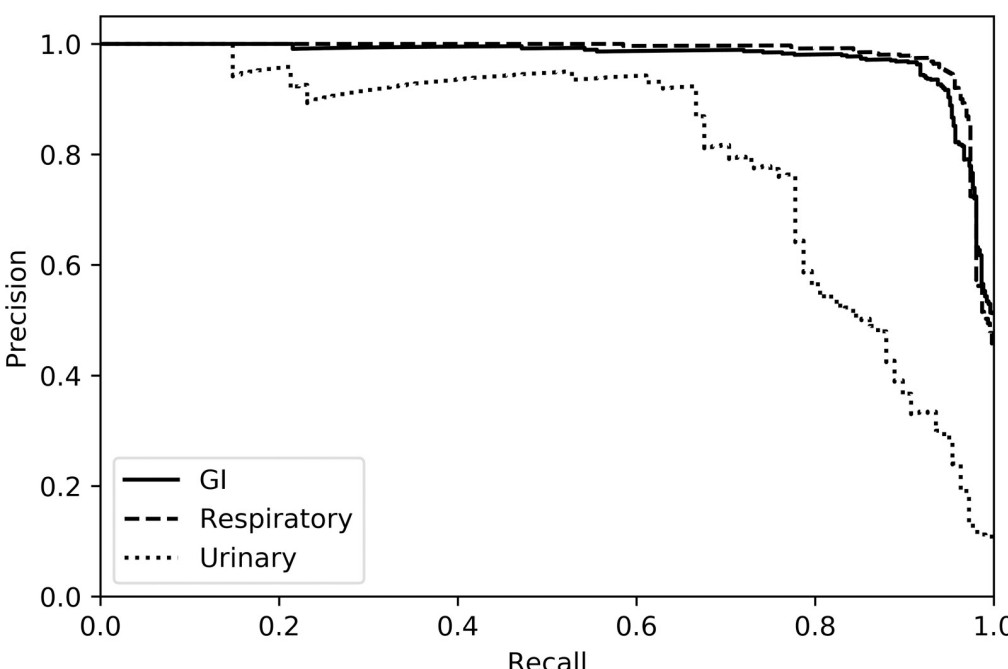

**Fig 6. Precision-recall curves for lstm network.** Areas under the curve (AUCs) are 0.975 (GI), 0.982 (respiratory), and 0.796 (urinary).

contained terms in more than one category. Table D in S1 Text presents examples of features encountered in this error analysis.

Among the false positives, 81% (43/53) were associated with text that mentioned a biological entity without suggesting any corresponding pathology. In such cases, the pathology report may state that a specific organ or tissue has no lesions or may describe findings associated with normal postmortem processes. Furthermore, 87% (46/53) of all false positive predictions were associated with documents for which the original report's *morphologic findings* and *final*

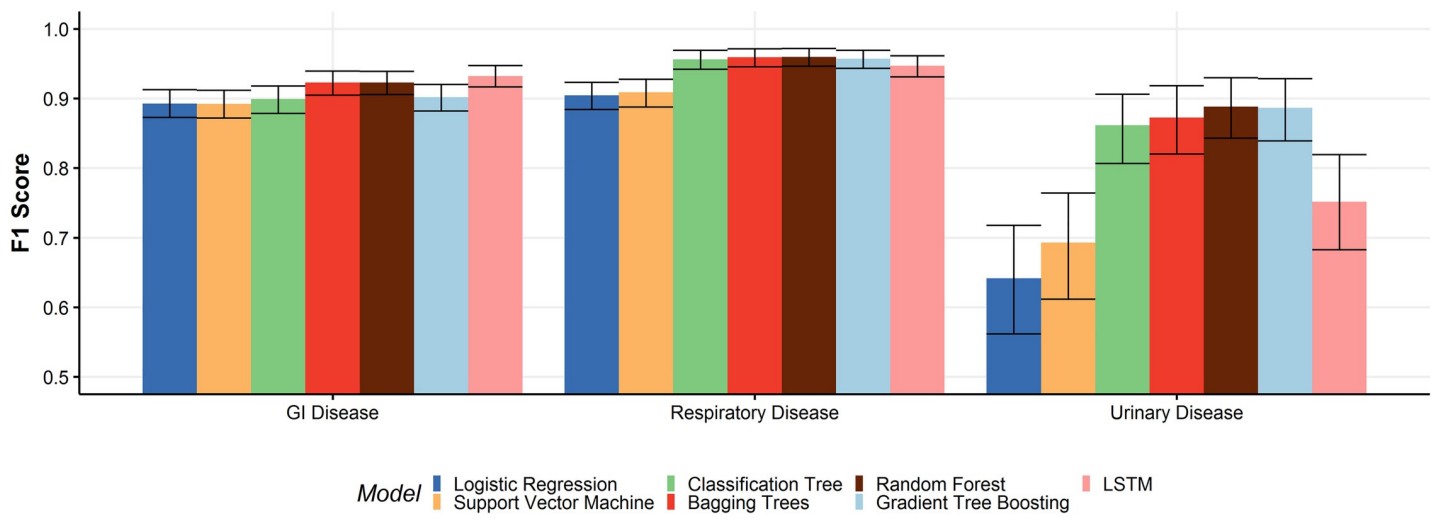

**Fig 7. Comparison of F1 scores for all models.** F1 scores were calculated using 10-fold cross-validation and bootstrapping to produce 95% confidence intervals (error bars).

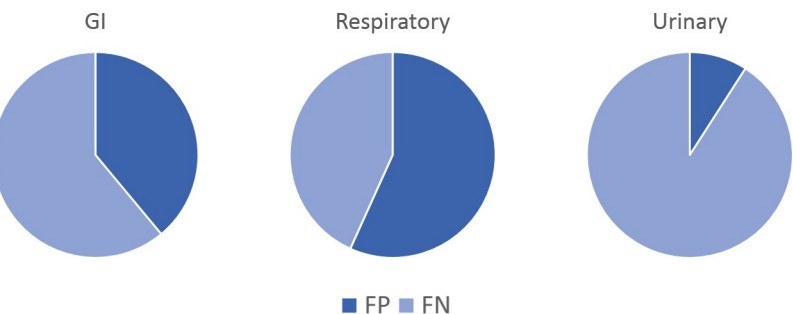

**Fig 8. Error rates of the random forest model.** Percentages of error due to false negatives and false positives.

*diagnosis* sections were both empty (and therefore the remaining sections were used for training). This is noteworthy because only 15% of original reports fall into this atypical group.

## Classifying documents beyond the labeled corpus

A random forest was used to render syndromic predictions on the entire document corpus with hyperparameters $m = 0.1p$ (for feature space of size $p$) and $min_{samples} = 2$ selected by 10-fold cross-validated grid search. Distributions of predicted monthly GI-disease counts for necropsy cases at the Wisconsin Veterinary Diagnostic Laboratory (WVDL) are illustrated in Fig 9.

The random forest predictions were used to generate a time-series of GI disease counts among species labeled as "small animal exotic" (Fig 10). An apparent increase in GI disease

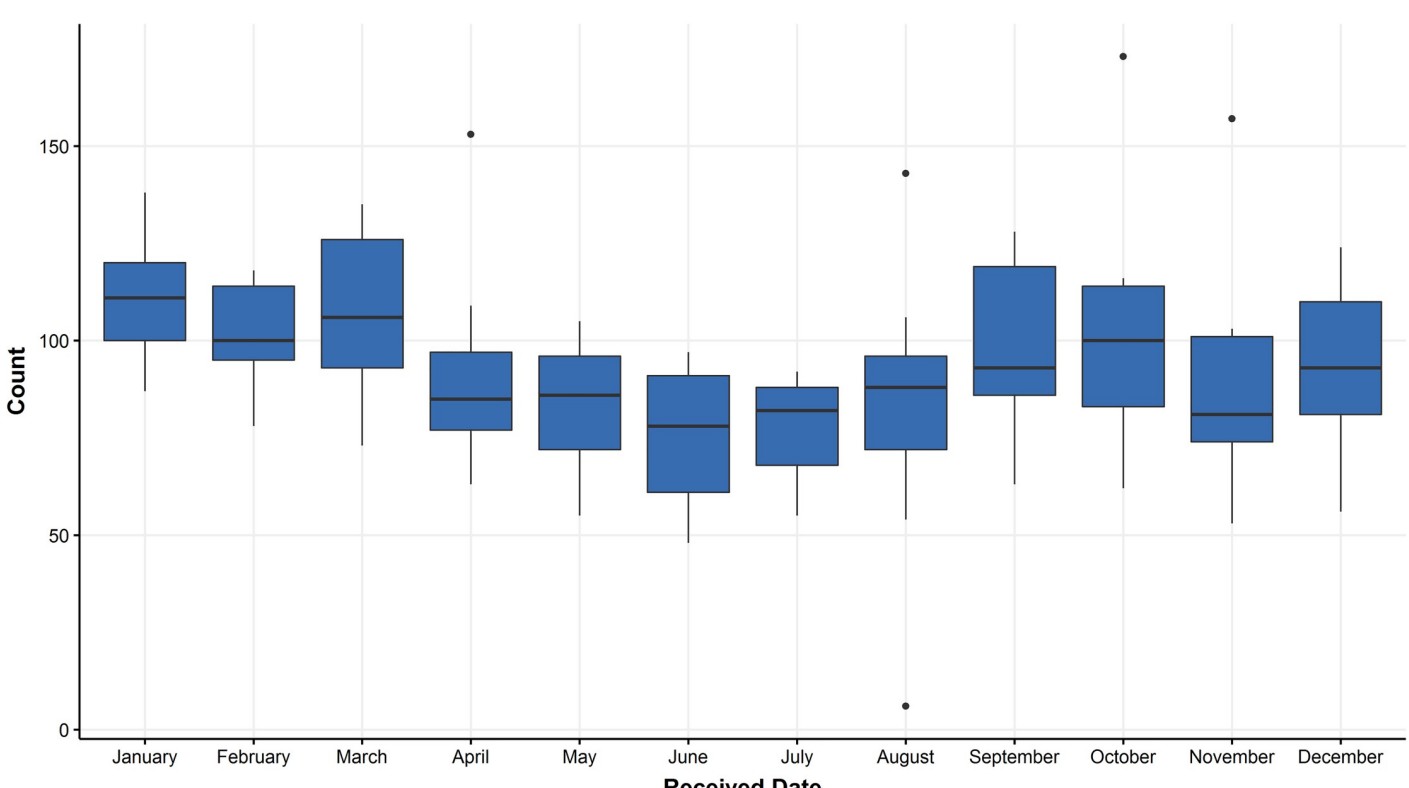

**Fig 9. Monthly counts of GI syndrome.** Each boxplot depicts a distribution of the total number of GI cases within a calendar month using syndromic predictions from 14 years of WVDL necropsy reports.

was observed in the fall of 2016. Further examination of cases contributing to this phenomenon revealed that many specimens came from a single producer, and their necropsy reports included evidence of non-specific hepatic pathology.

### Learning from gross necropsy findings

We measured the predictive signal represented by the first section of necropsy reports (*gross necropsy findings*), which are often written earlier than other sections. There were 622 labeled documents with non-empty *gross necropsy findings*, and the syndrome prevalences within this subset were 0.532 (GI), 0.461 (respiratory), and 0.140 (urinary).

When models were tested on gross findings, F1 scores of 0.738 (GI), 0.698 (respiratory), and 0.423 (urinary) were achieved by a support vector machine, random forest, and classification tree respectively (Table 5). For GI and respiratory disease, the most performant models were trained on gross findings. While several learners achieved F1 scores exceeding the baseline classifier, no models outperformed it with statistical significance on the GI or respiratory disease tasks. For urinary disease, classification tree and bagging trees models trained on primary documents outperformed the baseline classifier with statistical significance.

## Discussion

### Algorithm performance

This study demonstrates that it is feasible to use machine learning algorithms to classify veterinary necropsy reports according to their mention of GI, respiratory, or urinary disease. The F1

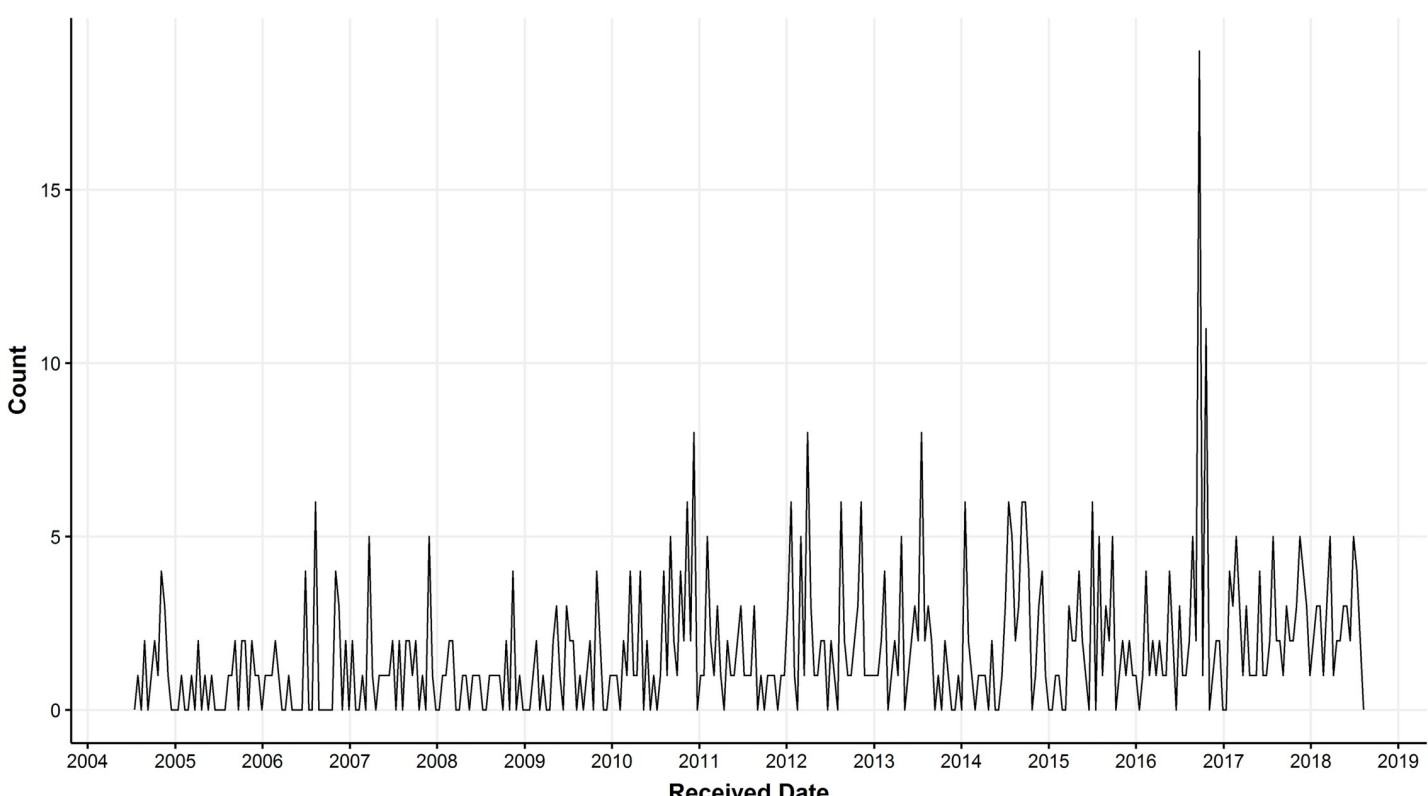

**Fig 10. Time series of GI cases.** Number of GI cases in small animal exotic species, based on syndromic predictions. There is a noteworthy increase in the number of GI cases in the fall of 2016. Counts are based on grouping cases into 14-day bins.

**Table 5. F1 Scores for models trained on primary documents or gross findings and tested on gross findings.**

| Model | GI Disease Primary Documents | GI Disease Gross Findings | Respiratory Disease Primary Documents | Respiratory Disease Gross Findings | Urinary Disease Primary Documents | Urinary Disease Gross Findings |
|---|---|---|---|---|---|---|
| **Logistic Regression** | 0.579 (0.528, 0.628) | **0.731** **(0.694, 0.764)** | 0.474 (0.417, 0.529) | 0.657 (0.610, 0.699) | 0.268 (0.165, 0.368) | 0.255 (0.159, 0.350) |
| **Support Vector Machine** | 0.536 (0.485, 0.587) | **0.738** **(0.700, 0.773)** | 0.409 (0.350, 0.471) | 0.642 (0.597, 0.685) | 0.254 (0.154, 0.359) | 0.247 (0.157, 0.343) |
| **Classification Tree** | 0.617 (0.570, 0.663) | 0.646 (0.604, 0.686) | 0.619 (0.569, 0.669) | 0.626 (0.578, 0.671) | **0.423** **(0.324, 0.516)** | 0.263 (0.175, 0.355) |
| **Bagging Trees** | 0.598 (0.547, 0.642) | 0.720 (0.678, 0.756) | 0.599 (0.546, 0.652) | **0.692** **(0.648, 0.734)** | **0.394** **(0.294, 0.491)** | 0.041 (0.000, 0.104) |
| **Random Forest** | 0.574 (0.522, 0.621) | 0.714 (0.673, 0.749) | 0.606 (0.555, 0.654) | **0.698** **(0.654, 0.743)** | 0.374 (0.277, 0.471) | 0.042 (0.000, 0.104) |
| **Gradient Tree Boosting** | 0.590 (0.537, 0.635) | 0.713 (0.673, 0.752) | 0.597 (0.545, 0.647) | 0.634 (0.586, 0.681) | 0.372 (0.271, 0.469) | 0.230 (0.145, 0.319) |
| **Baseline Classifier** | 0.695 (0.661, 0.725) | | 0.631 (0.594, 0.668) | | 0.245 (0.202, 0.287) | |

Models were trained with primary documents or the *gross necropsy findings* section using TF-IDF feature representations. Machine learning classifiers were tested on gross findings, and a baseline classifier produced a positive prediction on every document. F1 scores are reported here with 95% confidence intervals shown in parentheses. Within each syndrome, the two best results are bolded, and shaded results are superior to the baseline classifier with statistical significance.

scores are high and models showed significant levels of recall at high rates of precision. The best-performing algorithms are at least comparable to models performing other information-extraction tasks on free-text pathology reports, where micro F1 scores are often reported in the 0.45–0.92 range [24,29,35]. No machine learning algorithm outperformed all others by a statistically significant margin, although the random forest algorithm had consistently high performance across all three syndromic prediction tasks.

Error analysis of the random forest model suggests that higher performance may be possible if we strengthen its ability to infer correct syndromic labels from important but rare biomedical terms. This issue might be addressed using medical ontologies such as the Unified Medical Language System® (UMLS®) from the U.S. National Library of Medicine (NLM) to link conceptually related medical terms. In the future, machine learning systems for syndromic surveillance may use such frameworks to make intelligent predictions from features that appear infrequently or which may be absent from training documents, as has been previously suggested in the context of rule-based syndromic classifiers [49].

Performance across the syndromic categories was variable, with GI and respiratory diseases being easier syndromes to detect as measured by average F1 scores across models. The urinary models had lower F1 scores due to poor recall, as depicted in the precision-recall curves and reflected in the finding that random forest false negatives outweigh false positives. During manual inspection of random forest predictions, it was also found that more prevalent features of urinary disease are associated more strongly with true positive predictions (e.g., terms such as "nephritis", "nephrosis", and "tubules"). These results suggest that errors in urinary syndrome prediction may arise due to the low prevalence of urinary disease in labeled documents. Non-random sampling methods such as the synthetic minority over-sampling technique (SMOTE) could help create a class-balanced dataset more appropriate for training a machine learning system for this task [50].

False positives in the random forest model were associated with documents that included specific biological terms without conferring a pathologic diagnosis, such as in a statement of negation (e.g., "Kidney: No significant lesions found."). False positives were also associated

with documents for which the original report did not contain a morphologic or final diagnosis, which most often resulted in a longer description of gross findings being used as input for the learning system. This suggests that statements of negation and longer texts (which are more likely to contain such statements) elevate the risk of false positives in this system.

By taking whole documents as input, recurrent neural network models like the LSTM network can learn to distinguish variations in sentence structure including statements of negation that become problematic when text is tokenized into unigram TF-IDF statistics. In this study, LSTM performance did not exceed the best-performing TF-IDF feature-vector models for GI and respiratory disease and was markedly lower for urinary disease. Therefore, despite the theoretical advantages of recurrent neural networks and the recent evidence that they are effective for chief complaint classification in human medicine [10], in this domain we were unable to conclude that they are superior to models using unigram TF-IDF feature representations. Like most deep learning algorithms, LSTM networks often require very large datasets to train effectively. It is possible that deep learning could outperform TF-IDF feature-vector approaches with more training input, but with our relatively small dataset of 1,000 necropsy reports it was not possible to test this hypothesis. In future studies, active learning algorithms may help guide the document-labeling process to ensure that limited training data is optimally informative.

## Syndromic surveillance

After the performance of a machine learning classifier has been validated, it can be applied to a larger collection of historical data and its syndromic predictions can be leveraged to draw epidemiological conclusions. Using predictions from the random forest algorithm, we can track syndromes over time and localize cases involved in a suspected outbreak. Our example in Fig 10 illustrated increased GI disease in animals originating from locations close to the diagnostic lab. This approach could also help uncover baseline trends in case numbers at this laboratory. Analysis of historical trends is valuable for resource planning at a diagnostic laboratory and may help test hypotheses about important diseases in the case population.

Studies conducting further analysis of syndromic time series could generate real-time syndromic surveillance applications. For example, future studies may consider a hierarchical surveillance pipeline in which syndromic predictions are processed using statistical event-detection algorithms to detect emerging anomalies in real time.

In a standard necropsy workflow, it is common for a pathologist to begin by describing gross findings and then to state morphologic and final diagnoses after histological evaluation of tissues and ancillary lab testing. In this study, primary documents were prepared by taking only *morphologic findings* and *final diagnosis* sections from necropsy reports, in cases where at least one of these sections was non-empty. This means that training material may have included findings influenced by laboratory tests. However, in real-time use cases, a syndromic surveillance application would ideally make preliminary predictions on initial drafts of reports before test results are available and would update its predictions as reports are completed.

Our findings suggest that gross necropsy text on its own presents a weak predictive signal for syndromic surveillance. There are several reasons for this weakness. First, gross findings can be subtle or non-specific in some disease conditions. Second, 38% of labeled documents did not have information populated in the *gross necropsy findings* section. Third, we assumed that the label assigned to the primary document logically transfers to the gross findings. This assumption does not hold true in cases where the first section contains a statement of non-diagnostic significance such as "see below". Finally, error analysis showed that a high proportion of false positives were associated with longer documents, and gross findings often represent a longer narrative-style text. Future development of necropsy syndromic surveillance

applications should consider the informativeness of initial drafts or gross findings when supported by the medical database system. Institutional policies that may address this need include guidelines for uniform usage of each necropsy report section or version control systems that would enable direct analysis of initial drafts.

Syndromic surveillance of animal populations can provide epidemiological insights that are important to animal and public health. While structured medical data (such as records that include coded diagnoses) would simplify the design of syndromic surveillance systems, there is still an abundance of free-text data in veterinary medicine. The methods presented in this paper provide a framework for extracting syndromic information from free-text necropsy reports. Machine learning approaches may also help to automate veterinary syndromic surveillance using other types of medical text, including physical exam findings and discharge documents. Further work may examine the utility of machine learning for veterinary syndromic surveillance in these domains.

## Supporting information

**S1 Text. Supplemental tables.**
(DOCX)

## Acknowledgments

We are grateful to Dr. Tony Goldberg and Dr. Marie Pinkerton at the University of Wisconsin School of Veterinary Medicine for their mentorship and advice. We would like to thank Dr. Daniel Walsh at the National Wildlife Health Center for his insight regarding hierarchical surveillance pipelines. The authors also appreciate the support of Dr. Keith Poulsen and the IT assistance of David LaBeause at the Wisconsin Veterinary Diagnostic Laboratory. Finally, many thanks to Dr. David Page for his encouragement and guidance.

## Author Contributions

**Conceptualization:** Nathan Bollig, Mark Craven.

**Data curation:** Nathan Bollig, Lorelei Clarke, Elizabeth Elsmo.

**Formal analysis:** Nathan Bollig.

**Investigation:** Nathan Bollig.

**Methodology:** Nathan Bollig.

**Project administration:** Nathan Bollig.

**Resources:** Elizabeth Elsmo.

**Software:** Nathan Bollig.

**Supervision:** Mark Craven.

**Validation:** Nathan Bollig.

**Visualization:** Nathan Bollig.

**Writing – original draft:** Nathan Bollig.

**Writing – review & editing:** Nathan Bollig, Lorelei Clarke, Elizabeth Elsmo, Mark Craven.

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
