## [Decision Letter · Decision Letter 0]

18 Oct 2019

PONE-D-19-21922

Machine learning for syndromic surveillance using veterinary necropsy reports

PLOS ONE

Dear Dr. Bollig,

Thank you for submitting your manuscript to PLOS ONE. After careful consideration, we feel that it has merit but does not fully meet PLOS ONE’s publication criteria as it currently stands. Therefore, we invite you to submit a revised version of the manuscript that addresses the points raised during the review process.

Please correct the manuscript according to all reviewers 'comments and answer all reviewers' comments point by point.

We would appreciate receiving your revised manuscript by Dec 02 2019 11:59PM. To enhance the reproducibility of your results, we recommend that if applicable you deposit your laboratory protocols in protocols.io, where a protocol can be assigned its own identifier (DOI) such that it can be cited independently in the future. For instructions see: http://journals.plos.org/plosone/s/submission-guidelines#loc-laboratory-protocols

We look forward to receiving your revised manuscript.

Kind regards,

Paweł Pławiak, Ph.D.

Academic Editor

PLOS ONE

Journal Requirements:

2. We note that Figure 8 in your submission contains [map/satellite] images which may be copyrighted. All PLOS content is published under the Creative Commons Attribution License (CC BY 4.0), which means that the manuscript, images, and Supporting Information files will be freely available online, and any third party is permitted to access, download, copy, distribute, and use these materials in any way, even commercially, with proper attribution. For these reasons, we cannot publish previously copyrighted maps or satellite images created using proprietary data, such as Google software (Google Maps, Street View, and Earth). For more information, see our copyright guidelines: http://journals.plos.org/plosone/s/licenses-and-copyright.

1.    You may seek permission from the original copyright holder of Figure 8 to publish the content specifically under the CC BY 4.0 license.  

Additional Editor Comments:

Please correct the manuscript according to all reviewers 'comments and answer all reviewers' comments point by point.

Reviewers' comments:

Reviewer's Responses to Questions

**Comments to the Author**

1. Is the manuscript technically sound, and do the data support the conclusions?

Reviewer #1: Yes

Reviewer #2: Yes

2. Has the statistical analysis been performed appropriately and rigorously? 

Reviewer #1: Yes

Reviewer #2: Yes

3. Have the authors made all data underlying the findings in their manuscript fully available?

Reviewer #1: No

Reviewer #2: No

4. Is the manuscript presented in an intelligible fashion and written in standard English?

Reviewer #1: Yes

Reviewer #2: Yes

5. Review Comments to the Author

Reviewer #1: Overall, this is an exciting and well-written paper. The paper presents a novel task of classifying necropsy reports from the Wisconsin Veterinary Diagnostic Laboratory into one of three classes: Gastrointestinal Disease, Respiratory Disease, or Urinary Disease. One thousand examples are annotated, and the Cohen's kappa is recorded at 0.944, suggesting almost perfect agreement. Finally, multiple methods are compared, including, but not limited to, SVM, Logistic Regression, Random Forest, and an LSTM. Moreover, they perform interesting error analysis and provide possible policies to overcome certain errors.

Major Comments:

The exact counts for each of the classes should be specified, not just the percentage in the dataset.

The bootstrapped cross-validation procedure should be described in the manuscript. It is not a standard methodolgy that readers may not be familiar with.

The use of unigrams limits the performance of the Logistic Regression and the SVM models. I believe the Logistic Regression model---or an SVM with a linear kernel (check LinearSVC in sklearn)---would perform better if both tfidf-weighted unigrams and bigrams are included.

The LSTM results should be included in Table 1. Otherwise, it is cumbersome to compare Table 4 with Table 2.

Minor Comments:

While out of scope for this paper, it would be interesting to know if pretraining neural network-based models on non-veternarian electronic medical records (e.g., MIMIC) could improve its performance.

The data availability statement should be mentioned in the manuscript.

Reviewer #2: This work is a comparative study of machine learning algorithms used to detect if a necropsy report from the Wisconsin Veterinary Diagnostic 25 Laboratory contains evidence of gastrointestinal, respiratory, or urinary pathology. No novel approaches are presented. The statistical analysis has been conducted in a proper manner and a thorough discussion was presented. The manuscript is well written and the results are clearly stated.

Some minor comments:

Lines 95,96 combining the morphologic findings and final diagnosis sections or, if both of those were empty, by combining all sections (15% of cases).

Have you tried any other ways of dealing with empty data elements?

Line 109 “Phrases from Wisconsin Veterinary Diagnostic Laboratory (WVDL) necropsy reports that were judged by a human reviewer as representing gastrointestinal, respiratory, or urinary pathology.”

Perhaps more than one human reviewer should provide the judgements.

Line 113 “Two veterinarians board-certified by the American College of Veterinary Pathologists reviewed the 1,000 documents and classified each as having evidence of GI disease and/or respiratory disease and/or urinary disease based on clinical experience. “

Having more than two veterinarians involved in the study would result in a more reliable review.

Line 438 “This approach could also help uncover baseline trends in case numbers. “

Line 77 “…automated approach to syndromic 78 surveillance within animal populations.”

The data used in the study was obtained from one center: Wisconsin Veterinary Diagnostic Laboratory (WVDL) at the 86 University of Wisconsin-Madison.

The use of data from one center in the case of generalization suggests that there may exist certain trends in the way the necropsy reports are prepared by veterinarians. Using data from more than one veterinary laboratory would provide a more appropriate basis for generalizing the results.

6. PLOS authors have the option to publish the peer review history of their article (what does this mean?). If published, this will include your full peer review and any attached files.

Reviewer #1: No

Reviewer #2: No

---

## [Author Response · Author response to Decision Letter 0]

29 Nov 2019

We discuss all reviewer comments point by point in the Response to Reviewers letter.

---

## [Decision Letter · Decision Letter 1]

8 Jan 2020

Machine learning for syndromic surveillance using veterinary necropsy reports

PONE-D-19-21922R1

Dear Dr. Bollig,

We are pleased to inform you that your manuscript has been judged scientifically suitable for publication and will be formally accepted for publication once it complies with all outstanding technical requirements.

With kind regards,

Paweł Pławiak, Ph.D.

Academic Editor

PLOS ONE

Additional Editor Comments (optional):

Reviewers' comments:

Reviewer's Responses to Questions

**Comments to the Author**

1. If the authors have adequately addressed your comments raised in a previous round of review and you feel that this manuscript is now acceptable for publication, you may indicate that here to bypass the “Comments to the Author” section, enter your conflict of interest statement in the “Confidential to Editor” section, and submit your "Accept" recommendation.

Reviewer #1: All comments have been addressed

2. Is the manuscript technically sound, and do the data support the conclusions?

Reviewer #1: Yes

3. Has the statistical analysis been performed appropriately and rigorously? 

Reviewer #1: Yes

4. Have the authors made all data underlying the findings in their manuscript fully available?

Reviewer #1: Yes

5. Is the manuscript presented in an intelligible fashion and written in standard English?

Reviewer #1: Yes

6. Review Comments to the Author

Reviewer #1: (No Response)

7. PLOS authors have the option to publish the peer review history of their article (what does this mean?). If published, this will include your full peer review and any attached files.

Reviewer #1: No

---

## [Editor Report · Acceptance letter]

28 Jan 2020

PONE-D-19-21922R1 

Machine learning for syndromic surveillance using veterinary necropsy reports 

Dear Dr. Bollig:

I am pleased to inform you that your manuscript has been deemed suitable for publication in PLOS ONE. Congratulations! Your manuscript is now with our production department. 

With kind regards,

on behalf of

Dr. Paweł Pławiak 

Academic Editor

PLOS ONE